# Genomic characterization of clinical and environmental isolates of *Bacillus anthracis* in Peru

Eissen E. Guerrero-Seminario[1,2], Abraham Espinoza-Culupú[1], Patricia García-Vara[2], John Calderón-Escalante[2], Dana González-Quispe[2], Ever F. Córdova-Diaz[2], Beitzy Cubas-Yalle[2], M. Angelica Delgado-Baldeon[2], Lourdes Balda Juárez[2], Ruth García-de-la-Guarda[1]*

1 Molecular Microbiology and Biotechnology Laboratory, Faculty of Biological Sciences, Universidad Nacional Mayor de San Marcos, Lima, Peru, 2 Laboratorio de Referencia Nacional de Metaxénicas y Zoonosis Bacterianas, Centro Nacional de Salud Pública, Instituto Nacional de Salud, Lima, Peru

* rgarciad@unmsm.edu.pe

## Abstract

Anthrax, or charbon, is a zoonotic disease caused by the Gram-positive bacterium *Bacillus anthracis*, responsible for sporadic outbreaks and representing a significant public health issue in Peru. In this study, 18 isolates from humans, animals and soil, collected between 2005 and 2017 across different country regions, were reactivated and analyzed. We identified antimicrobial resistance genes, virulence factors, plasmids, and their phylogenetic relationships using next-generation sequencing and bioinformatics tools. We detected eight genes associated with multidrug resistance, including *vanZ-F*, *mphL*, *fosB_gen*, *fosB*, *satA*, *bla*, *bla2*, and *fosB2*. Additionally, 24 chromosomal virulence genes were identified, related to toxin production, capsule biosynthesis, secretion systems, and iron acquisition. Analysis of the pXO1 plasmid revealed the presence of 67 virulence-related genes, such as *pagA*, *lef*, *cya*, *atxA*, and *pagR*, while the pXO2 plasmid exhibited 24 genes, highlighting the *cap* genes associated with capsule synthesis. Phylogenetic analysis revealed that all strains belong to group A, subgroup A.Br.003/004, providing valuable evidence about the evolutionary dynamics and genomic diversity of *B. anthracis* in Peru. These findings enhance our understanding of the genetic mechanisms involved in antimicrobial resistance and virulence, offering crucial insights for epidemiological surveillance and outbreak control.

## Author summary

Anthrax, or charbon, is a severe disease that causes sporadic outbreaks in endemic regions of northern and central Peru, affecting both animals and humans. This study presents the first genomic characterization analysis conducted in

**Data availability statement:** Illumina raw reads of isolates have been submitted to NCBI under BioProject PRJNA1267694.

**Funding:** The workstation used for computational genomic analyses was acquired through funding provided by internal research by Vicerrectorado de Investigación y Posgrado de la Universidad Nacional Mayor de San Marcos, grant number B2410003m. The funders had no role in study design, data collection and analysis, decision to publish, or preparation of the manuscript. Furthermore, no authors received a salary from this funder.

**Competing interests:** The authors have declared that no competing interests exist.

Peru, based on isolates collected between 2005 and 2017. It provides valuable insights into the genetic relationships between isolates, clinically significant antimicrobial resistance genes, and virulence genes, suggesting the presence of multiple pathogenicity mechanisms. In this study, 18 isolates were reactivated and sequenced to genetically characterise Peruvian strains of *Bacillus anthracis*. Using bioinformatics tools, we found that all isolates exhibited high genetic similarity. Additionally, genes associated with resistance to certain antibiotics, important in public health and genes involved in toxin production and immune system evasion were identified. This summary highlights key findings that contribute to understanding the genetic mechanisms underpinning the pathogenicity and resistance of *B. anthracis* in Peru, providing a foundation for improved epidemiological surveillance and disease control strategies.

## Introduction

Anthrax or charbon is a zoonotic disease caused by *Bacillus anthracis*, a Gram-positive, spore-forming, facultative aerobic bacterium capable of producing a capsule [1]. This disease primarily affects herbivores due to their constant exposure to environmental spores. However, other mammals, including humans, can become infected, particularly through ingesting or handling infected animals. The severity in humans depends on the route of exposure, which can be cutaneous, gastrointestinal, or pulmonary [2]. To date, human-to-human transmission has not been confirmed, although infection through direct contact with cutaneous lesions or tissues of infected animals has been documented [3].

The potential use of anthrax as a biological weapon was identified during World War II due to its spore resilience, ease of distribution and production, and high capacity to cause harm [4]. Despite the availability of effective treatments and their global distribution, anthrax outbreaks continue to occur sporadically in various regions worldwide [5].

In Peru, records of anthrax date back to the early 20th century, with more frequent reports starting in 1990, particularly in the coastal regions of the country's central and northern areas [6]. Between 1969 and 2002, a retrospective study in Lima reported 71 cases of cutaneous anthrax, including three fatalities [7]. A 2015 outbreak in northern Peru identified 44 human cases presenting with classic symptoms of the disease [8]. According to official records from the Ministry of Health of Peru (Ministerio de Salud, MINSA) [9], a total of 412 anthrax cases were reported between 2000 and 2024, with historically higher incidence in the departments of Ica, Piura, and Lima. Following an absence of reported cases between 2021 and 2024, an emerging outbreak was detected in the Cerro Azul district (Lima), with 6 confirmed human cases up to epidemiological week 40 of 2025 [10]. This event is associated with direct exposure to infected cattle and the handling of animal products, with the presence of *B. anthracis* confirmed in livestock remains and butchery utensils. In response, the

National Agrarian Health Service of Peru (Servicio Nacional de Sanidad Agraria, SENASA) [11] implemented an official vaccination campaign across nine coastal departments—from Tumbes to Tacna—aimed at reducing animal prevalence and minimizing the risk of zoonotic transmission.

The clinical manifestations of anthrax occur through three main routes of infection: cutaneous, accounting for 95% of human cases; gastrointestinal; and inhalational, which are less common [12]. In *Bacillus anthracis*, the genome structure consists of a single circular chromosome of 5,227,293 bp [13]. Virulence is conferred by two plasmids: pX01 (182 kb) and pX02 (96 kb). The pX01 plasmid encodes binary toxins, including lethal factor (LF) and edema factor (EF), which are calmodulin-dependent, as well as the adhesion subunit known as the protective antigen (PA), which is shared by both factors. The pX02 plasmid plays a critical role in macrophage survival and contains the capBCDAE operon, which is essential for the synthesis of the poly-γ-D-glutamic acid capsule (CAP) [14]. Without these plasmids, virulence is significantly reduced or lost [15]. Horizontal gene transfer within the Bacillus genus has been demonstrated under in vitro conditions [16].

*B. anthracis* exhibits remarkable genetic stability due to its prolonged spore-phase survival. Phylogenetically, *B. anthracis* is divided into three clades: A, B, and C. Clade A has a cosmopolitan distribution and comprises four monophyletic subclades. Clades B (10% of global strains) and C are more geographically restricted [17]. Advanced genotyping techniques, such as whole-genome sequencing (WGS), provide enhanced capabilities for genome analysis and characterisation. Additionally, methods like canonical single-nucleotide polymorphisms (canSNP) are highly accurate for differentiating bacterial strains and establishing phylogenetic relationships [18,19].

In this study, we report for the first time the genomic characterisation of *Bacillus anthracis* isolates from Peru, analysing their genetic variability, virulence factors, and antimicrobial resistance

## Methods

### Ethics statement

The procedures of this study were carried out with the approval of the Ethics Committee of the Instituto Nacional de Salud, Lima, Peru, under the ID OT-017-23 and RD protocol No. 249-2023-DIIS/INS.

### Reactivation of strains

Eighteen selected strains were reactivated in duplicate using 5% blood agar and tryptic soy agar (TSA), incubated at 37°C for 24 hours [2]. Colonies displaying typical phenotypic characteristics of the microorganism were selected and subsequently replanted onto TSA agar. The entire process was conducted in a biosafety level 3 facility. The strains included in this study were isolated during technical assistance provided for epidemiological outbreaks in northern regions and Lima, Peru (Table 1 and Fig 1) and supplied by the National Reference Laboratory for Metaxenic and Zoonotic Bacterial Diseases at the National Institute of Health, Peru.

### DNA extraction and amplification

DNA was extracted from purified cultures using a commercial kit (Mini Kit QIAamp DNA, Qiagen, Germany) following the manufacturer's instructions. For genetic material amplification, specific primers targeting the pag gene of the *pXO1* plasmid were used, with the following sequences: PAG 67-5′ CAGAATCAAGTTCCCAGGGG ′3 and PAG 68-5′ TCG-GATAAGCTGCCACAAGG ′3 [20]. The reaction mixture for PCR consisted of 2 μL of each primer at 0.5 μM, 1.6 μL of dNTPs (10 mM), 5 μL of 1X PCR buffer, 0.6 μL of MgCl2 (1.5 mM), 9.4 μL of nuclease-free water, 0.4 μL of KAPA enzyme (0.02 U/μL), and 2 μL of extracted DNA. The amplification protocol included an initial denaturation at 94°C for 5 minutes, followed by 30 cycles of 94°C for 1 minute, 60°C for 1 minute, and 72°C for 1 minute, with a final extension at 72°C for 5 minutes [21].

**Table 1. List of reactivated *Bacillus anthracis* strains from 2005 to 2017 in Peru.**

| Strain | Region | Year of Isolation | Source |
|---|---|---|---|
| PER-INS-BA-01 | Lambayeque | 2014 | Animal |
| PER-INS-BA-02 | Piura | 2015 | Animal |
| PER-INS-BA-03 | Piura | 2015 | Animal |
| PER-INS-BA-04 | Cajamarca | 2016 | Soil |
| PER-INS-BA-05 | Lima | 2005 | Human |
| PER-INS-BA-06 | Lima | 2005 | Human |
| PER-INS-BA-07 | Lima | 2005 | Human |
| PER-INS-BA-08 | Lima | 2005 | Human |
| PER-INS-BA-09 | Lima | 2005 | Human |
| PER-INS-BA-10 | Lima | 2005 | Soil |
| PER-INS-BA-11 | Lima | 2005 | Human |
| PER-INS-BA-12 | Cajamarca | 2006 | Animal |
| PER-INS-BA-13 | Cajamarca | 2006 | Animal |
| PER-INS-BA-14 | Lambayeque | 2014 | Animal |
| PER-INS-BA-15 | Piura | 2017 | Animal |
| PER-INS-BA-16 | Piura | 2017 | Animal |
| PER-INS-BA-17 | Lima | 2005 | Human |
| PER-INS-BA-18 | Cajamarca | 2006 | Animal |

## Whole genome sequencing (WGS)

Whole genome sequencing was performed using the Illumina MiSeq platform (Illumina, USA) following the manufacturer's protocol. Genomic DNA was quantified using a fluorometer with the Qubit dsDNA HS Assay Kit (Thermo Fisher Scientific). Tagging (NTA) was performed using the Nextera XT DNA Library Prep Kit v3, which served as a 10 nM phiX control. Sequencing was performed using the MiSeq Reagent Kit v3 (Illumina, USA).

## Bioinformatics analysis

The reads obtained from sequencing were initially evaluated using FastQC [22] to assess their quality. Subsequently, adapters and low-quality sequences were removed using Trimmomatic [23]. The genomes were assembled de novo with SPAdes [24]. The resulting contigs were mapped against the reference genome of *Bacillus anthracis* Ames strain (NC_007530.2), as well as the pXO1 (NC_007322.2) and pXO2 (NC_007323.3) plasmids. Finally, the contigs were annotated using Prokka [25].

To identify antimicrobial resistance genes, the contigs were analyzed using the web-based platform of the Center for Genomic Epidemiology [26], specifically with ResFinder [27]. Additionally, AMRFinderPlus v3.12.8 with database version 2024-05-02.2 was used to detect resistance and virulence genes, complemented by analysis with VirulenceFinder [28]. Other bioinformatics platforms [29], such as the Virulence Factor Database and IslandPath [30], were also employed to identify virulence factors.

To evaluate the phylogenetic relationship between the Peruvian *Bacillus anthracis* strains and other representative strains worldwide, a core-genome alignment was performed using Parsnp v1.7.4, part of the Harvest Suite [31]. The Ames Ancestor strain was used as the reference genome, and the analysis included the complete genomes of the 18 Peruvian strains along with 35 international strains representative of various lineages and geographic regions, and the resulting tree was edited using iTOL [32]. In addition, a reference-based single nucleotide polymorphism (SNP) analysis was performed on the genomes listed in S1 Table using Snippy v4.6, with the Ames Ancestor genome as reference (ID: AE017334.2).

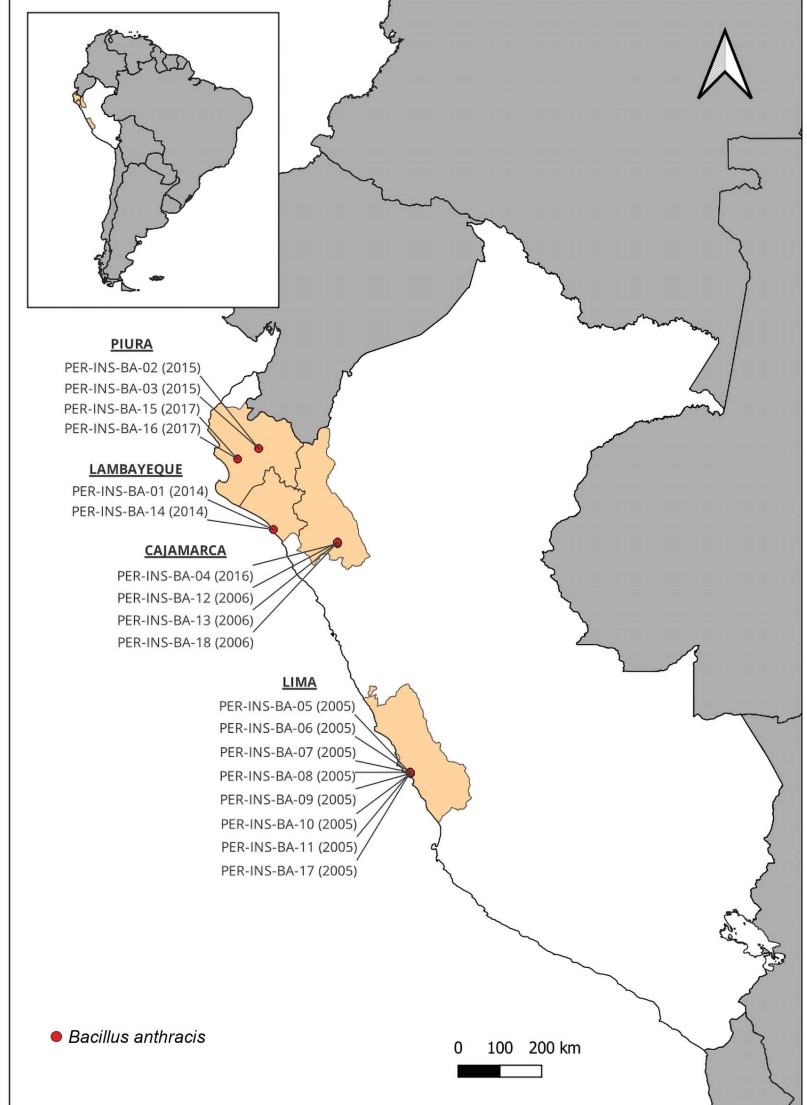

**Fig 1. Geographic distribution of *Bacillus anthracis* strains isolated in Peru from 2005 to 2017.** The map highlights regions of isolation, including Lambayeque, Piura, Cajamarca, and Lima. Strains were recovered from soil, animals, and human cases during epidemiological outbreak investigations. Reference departmental boundaries of Peru were obtained from the Peruvian National Open Data Platform (ODC-BY; https://www.datosabiertos.gob.pe/dataset/limites-departamentales), and country boundaries of South America were obtained from Natural Earth (public domain; https://www.naturalearth-data.com/downloads/110m-cultural-vectors/), compatible with CC BY 4.0.

Finally, the canSNP assignment for each strain was determined from the genome assemblies using CanSNPer2 [33] with the specific scheme available for *B. anthracis*.

All genomic analyses were conducted on a high-performance Lenovo ThinkStation P920 workstation equipped with an Intel Xeon Gold 6151 CPU and 128 GB of RAM, which was acquired through funding provided by internal research by Vicerrectorado de Investigación y Posgrado de la Universidad Nacional Mayor de San Marcos, grant number B2410003m

## Results

### Reactivation

Eighteen *Bacillus anthracis* strains from the culture collection were successfully reactivated and purified. These isolates were obtained from soil, animals, and human samples in the Peruvian regions of Lambayeque, Piura, Cajamarca, and Lima, during technical support provided for epidemiological outbreaks between 2005 and 2017 (Table 1 and Fig 1).

### Genomic identification of antimicrobial resistance genes

In the analysis of 18 *Bacillus anthracis* strains from different regions of Peru, eight genes associated with antimicrobial resistance were identified: *vanZ-F*, *mphL*, *fosB_gen*, *fosB*, *bla*, *satA*, *bla2*, and *fosB2* (Fig 2). These genes, detected in the majority of the analyzed genomes, indicate a multidrug resistance profile against various antibiotics. Furthermore, the genes showed an identity greater than 82%, highlighting the importance of their monitoring through genomic tools in future studies.

### Virulence and genetic diversity in Peruvian *Bacillus anthracis* strains

In the Peruvian strains of *Bacillus anthracis*, we identified 24 virulence genes with diverse functions, including toxin production, secretion systems, capsule biosynthesis, and iron acquisition, all exhibiting an identity greater than 94% (Fig 3).

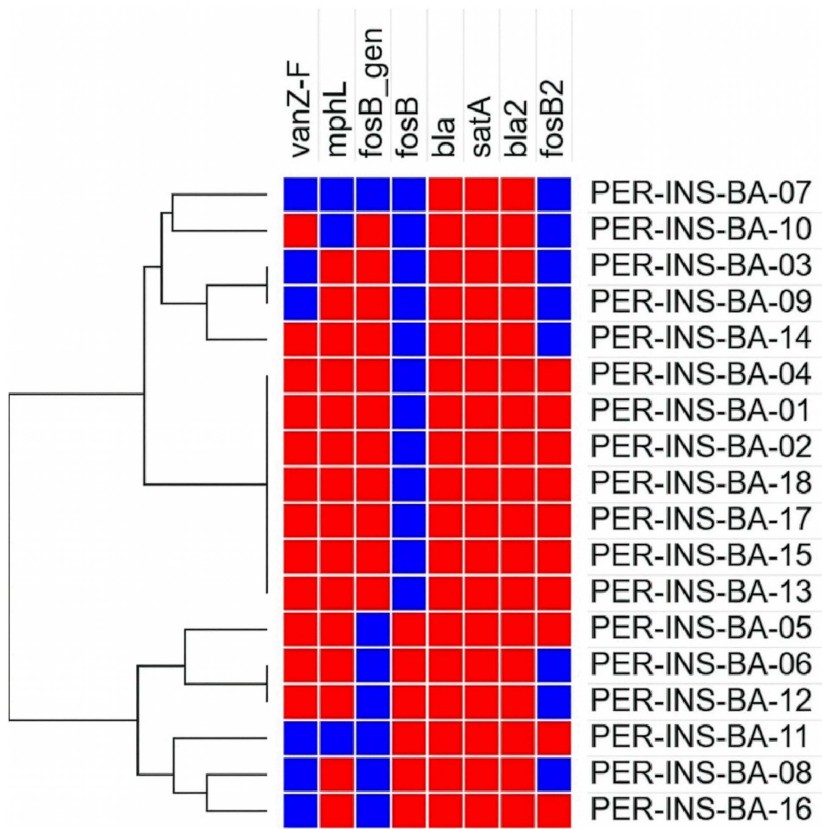

**Fig 2. Presence of antimicrobial resistance genes in Peruvian *Bacillus anthracis* strains.** The figure presents a heatmap displaying the antimicrobial resistance genes identified (*vanZ-F*, *mphL*, *fosB_gen*, *fosB*, *bla*, *satA*, *bla2*, *fosB2*) in 18 Peruvian *Bacillus anthracis* strains. Each row represents a strain, and each column indicates the presence (red) or absence (blue) of the genes. The dendrogram illustrates the similarity relationships among the strains based on the genes present.

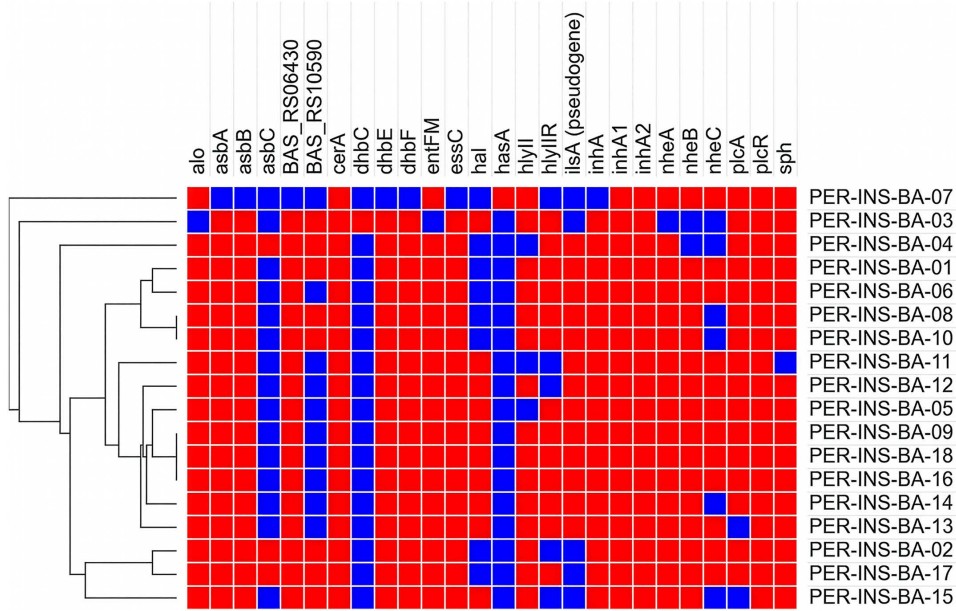

**Fig 3. Distribution of virulence genes in 18 Peruvian *Bacillus anthracis* strains.** The heatmap illustrates the presence/absence of 24 virulence genes identified in the analyzed strains, representing key functions such as toxin production, capsule biosynthesis, and iron acquisition. Red cells indicate the presence of genes with identity above 94%, while blue cells indicate their absence. The dendrogram on the left represents the hierarchical relationship among the strains based on their virulence profile, highlighting notable genetic conservation among the isolates.

## Genomic analysis of the pXO1 plasmid

The genomic analysis of the pXO1 plasmid revealed a remarkable conservation of key genetic elements associated with virulence, plasmid dynamics, and cellular processes. A total of 67 genes were identified, including three encoding the lethal anthrax toxin complex. Genes linked to genetic mobility and conjugative transfer, such as traG and repX, were detected alongside transposase elements from the IS3 and IS4 families. Additionally, the plasmid contains genes coding for spore germination proteins (*gerXA*, *gerXB*, *gerXC*) and transcriptional regulators from the TetR family. Furthermore, 175 proteins related to virulence, adaptation, genetic transfer, and stress response were identified. Among these, noteworthy elements include proteins with NEAT domains, integrases, HTH family transcriptional regulators (e.g., *MerR* and *ArsR*), RelA-SpoT proteins, and various transposases (Fig 4).

## Genomic analysis of the pXO2 plasmid

The analysis of the pXO2 plasmid identified 24 key genes involved in transcriptional regulation, cellular processes, and virulence. These include cap genes associated with capsule synthesis, regulators (*araC* and *toxI*), and genes essential for plasmid replication and maintenance (*reps* and *xerS*). Mobile elements such as transposases (tnp), a putative metal-binding protein (*yrpE*), and genes associated with sporulation and RNA processing (*spoVR*, *nusA*, *truB*, and *rpsO*) were also detected. Additionally, the plasmid harbors genes critical for chromosomal segregation (scpA and scpB), translation initiation (infB), and post-transcriptional regulation (*ylxR*, *ylxQ*, and *ylxP*). A total of 57 proteins with specialized functions were identified, providing insight into the pathogenic and regulatory capacity of the plasmid. These include intramembrane metalloproteases (CPBP family), DNA helicases, type VII secretion system proteins, and components with LXG, SIR2–2, and TPR-5 domains. Other notable elements include proteins involved in transposition, integration, transcriptional regulation (*HTH-TetR* and *TFIIS* domains), stress response, antitoxin systems (*ToxI*), metabolic enzymes such

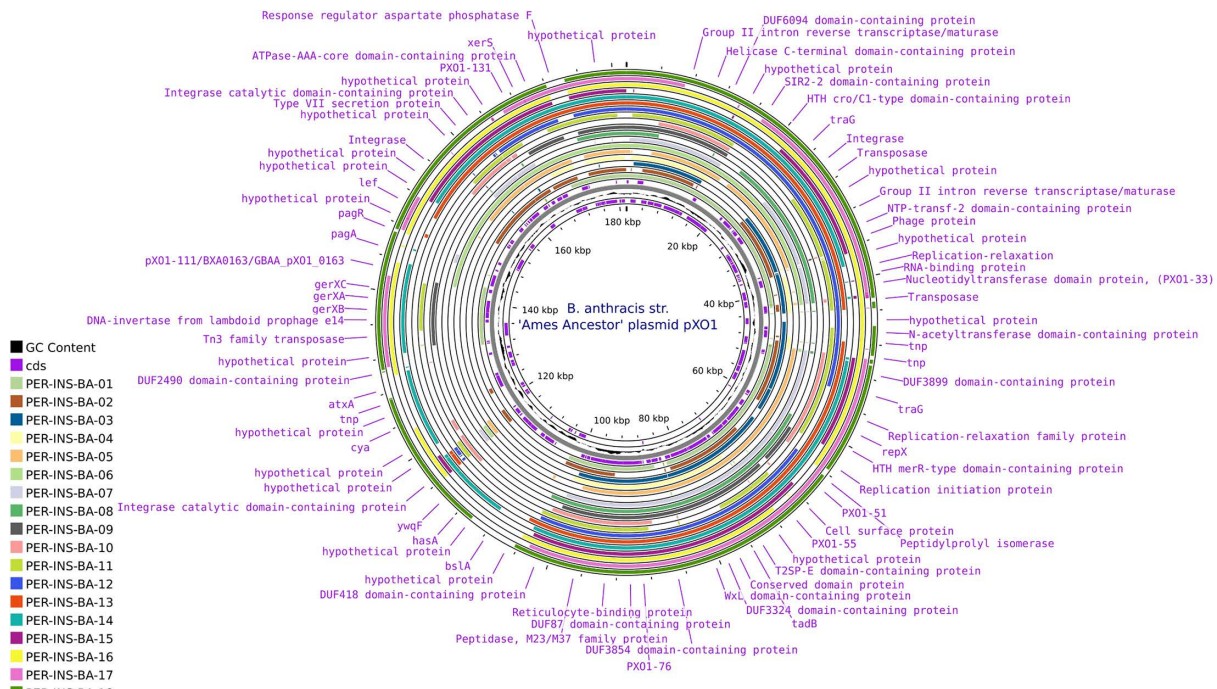

**Fig 4. Comparison of plasmid pXO1 in Peruvian *Bacillus anthracis* strains.** Circular representation of the pXO1 plasmid from the *B. anthracis* Ames Ancestor strain. Each ring corresponds to a Peruvian strain (PER-INS-BA-01 to PER-INS-BA-18). Gene names are displayed around the circle, including virulence factors, transposases, integrases, and hypothetical proteins. GC content (in black) and coding sequences (CDS, in purple) are also shown.

as ribonucleoside-diphosphate reductase, N-acetylglucosamine-6-phosphate deacetylase, chromosomal segregation/condensation proteins (*ScpA* and *ScpB*), and membrane transport.

## Phylogenetic analysis of Peruvian strain

The 18 *Bacillus anthracis* isolates from Peru, collected between 2005 and 2017 across five regions—Lima, Piura, Cajamarca, Lambayeque, and La Libertad—clustered within lineage A (Fig 5), forming a monophyletic clade corresponding to the A.Br.003/004 group. Most of the strains were recovered from human and animal cases, particularly in the Lima region (n = 7) during outbreaks reported in 2005. In contrast, isolates from Piura, Cajamarca, and Lambayeque were primarily obtained from animals and environmental sources (soil), suggesting sustained circulation in natural reservoirs. The temporal and geographical distribution, along with the high genomic similarity among these isolates, indicates that the A.Br.003/004 clade represents a stable and possibly endemic population within Peru. This limited diversity and phylogenetic clustering suggest that recent outbreaks may be linked to a common source or to strains persistently maintained in zoonotic reservoirs.

In addition, a reference-based SNP analysis was performed using Snippy v4.6 with the Ames Ancestor genome as reference. A summary of core genome SNP counts and variant types for each isolate is provided in S1 Table.

## Discussion

*Bacillus anthracis* is considered an enzootic bacterium in Peru [34], highlighting the relevance of this study as the first to genomically characterize Peruvian strains isolated between 2005 and 2017 through whole genome sequencing (WGS). A total of 18 strains from the central coastal and northeastern regions of the country were analyzed

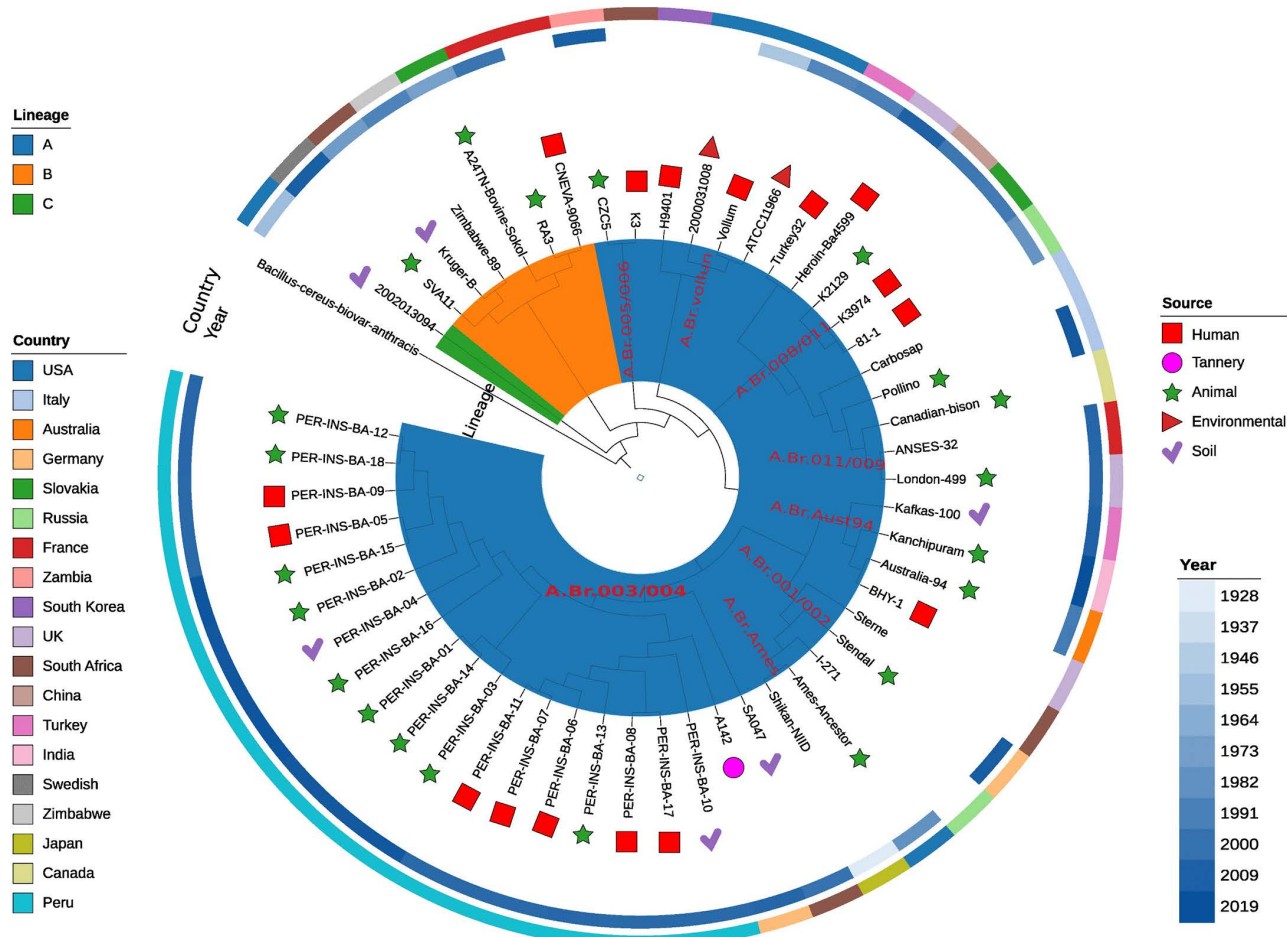

**Fig 5. Phylogenetic tree of *Bacillus anthracis* strains based on whole-genome sequences.** The circular tree displays the phylogenetic relationships among various *B. anthracis* strains worldwide. Isolates are grouped into three main lineages (A, B, and C), indicated by colors in the innermost ring. The Peruvian strains (n = 18) clustered within lineage A, specifically in the A.Br.003/004 group (highlighted in red). Outer symbols represent the source of isolation: human (red square), animal (green star), environmental (brown triangle), soil (purple heart), and tannery (pink circle).

(Table 1 and Fig 1), allowing for a comprehensive examination of the virulome, antibiotic resistance genes, and phylogenetic relationships. This information not only contributes to the enhancement of international databases but is also critical for understanding genetic relationships among strains and improving local systems for epidemiological surveillance and anthrax control.

The selection of WGS as an analytical tool is based on its high resolution and ability to provide robust genome characterization, as demonstrated in previous studies on the genetic, geographic, and historical diversity of *B. anthracis* [35]. In outbreak scenarios, this technology enables accurate identification and detailed comparison of strains [36]. In our study, Peruvian strains showed high similarity to the reference strain, with pairwise identity (PI) values above 97.9% in the chromosome, over 82% for antibiotic resistance genes (Fig 2), and over 94% for virulence genes (Fig 3). These values support the robustness of the WGS approach for genomic characterization of *B. anthracis*.

The notable genomic homogeneity observed among the analyzed strains is consistent with previous literature and may be attributed to the bacterium's prolonged latency periods in spore form. This survival strategy reduces the mutation rate and limits horizontal gene transfer events, thereby favoring genomic stability [17].

The analysis of antimicrobial resistance (AMR) genes in Peruvian *B. anthracis* strains revealed both similarities and differences compared to previous reports. Bruce et al. [37] frequently identified five genes—*bla1*, *bla2*, *fosB*, *vmlR*, and *mphL*—across multiple isolates. In contrast, our study identified a broader genetic diversity, with eight major AMR genes detected: *vanZ-F, mphL, fosB_gen, fosB2, fosB, bla, bla2*, and *satA*. Notably, three of these genes are associated with fosfomycin resistance, which may reflect differing selective pressures or regional variability in the bacterial population. The concordance in the high prevalence of *mphL* and *bla* genes across both studies supports their potential role in resistance to macrolides and beta-lactams, respectively. However, while *bla2* was detected in all strains in our study, Bruce et al. [37] reported a lower frequency, suggesting that geographic, temporal, and methodological factors may influence AMR gene distribution.

The detection of genes not previously reported, such as *vanZ-F* and *satA_Ba*, highlights the importance of continued genomic surveillance to identify emerging resistance mechanisms. Such findings have direct implications for both clinical management and epidemiological control of anthrax.

The widespread presence of *bla* and *bla2* among the isolates is particularly noteworthy given their association with resistance to beta-lactam antibiotics, which are standard treatments for anthrax. Nonetheless, the genomic presence of these genes does not necessarily imply phenotypic expression. Previous studies have demonstrated that *bla1* and *bla2* can confer resistance when expressed in Escherichia coli [38,39], and Gargis et al. [40] suggested that the sigP-bla1 locus may serve as a marker for penicillin resistance. However, the presence of AMR-associated mutations does not always correlate with a resistant phenotype. For instance, in an analysis of 40 Croatian isolates, all were susceptible to ciprofloxacin, penicillin, and tetracycline, despite harboring both *bla1* and *bla2* [41]. These findings underscore the importance of integrating genomic data with functional assays to accurately determine the clinical relevance of AMR gene carriage.

The virulome analysis revealed 34 genes associated with virulence factors, including toxins, secretion systems, capsule biosynthesis, and iron acquisition mechanisms, underscoring the notable genetic diversity present in Peruvian *B. anthracis* strains (Fig 3). These findings are consistent with those reported by Kompes et al. [41], who identified 21 virulence genes, 90% of which overlap with the present study. Genes such as *asbA*, *asbB*, *dhbE*, and *dhbF*, found in 94.4% of the strains, are involved in the biosynthesis of siderophores like petrobactin, a key molecule for iron acquisition and pathogenic expression [42–44]. Similarly, *essC* (94.4%) and *BAS_RS10590* (44.4%), both associated with the type VII secretion system (T7SS), suggest a potential role for this pathway in *B. anthracis* virulence [45].

The *inhA* gene in all strains encodes a metalloprotease implicated in tissue degradation and immune evasion mechanisms [46]. The high frequency of *alo* (94.4%) highlights the importance of anthralysin O as a cytotoxin contributing to cellular and tissue damage [47]. In contrast, *ilsA* was identified as a pseudogene in 66.6% of the isolates, potentially indicating a functional loss compensated by alternative iron uptake systems [48].

Although the *plcR* gene was detected in 94.4% of the strains, it is known to be inactivated in *B. anthracis* due to a point mutation, which accounts for its non-hemolytic phenotype [49,50]. However, other genes such as *plcA* (83.3%) and *sph* (94.4%) suggest the involvement of multiple factors in cellular lysis and immune evasion [51–53]. Additionally, *hlyII* and *hlyIIR* (>72.2%) are linked to pore formation and apoptosis induction in macrophages [54,55]. Lastly, the *nhe* gene cluster (>72.2%), which encodes a non-hemolytic enterotoxin, appears to play a less significant role in *B. anthracis* virulence compared to its function in *B. cereus* [56,57].

Analysis of the pXO1 plasmid revealed 67 virulence-associated genes, including the anthrax toxin complex (*pagA*, *lef*, *cya*) and key regulatory elements *atxA* and *pagR*, which are essential for the coordinated expression of toxins and capsule synthesis [58,59]. Genes involved in spore germination, such as *gerXA*, *gerXB*, and *gerXC*, were also identified [60]. In total, 175 functional proteins were annotated (Fig 4), including mobile genetic elements such as IS3 and IS4 family transposases, integrases, and HTH-type transcriptional regulators (e.g., *ArsR*, *MerR*), which may confer adaptive advantages under diverse environmental conditions [61–63].

The pXO2 plasmid analysis identified 24 core genes, including the *capBCADE* operon responsible for capsule biosynthesis, under the regulation of *acpA* and *acpB* [64,65]. Additional genes were found related to replication (*repS*), genetic mobility (*tnp*, *xerS*), and plasmid maintenance, such as *toxI*, part of a toxin-antitoxin system [61,66–68]. Fifty-five functional proteins were annotated, including mobile elements potentially involved in horizontal gene transfer [69].

Phylogenetic analysis placed the Peruvian strains within lineage A, sub-group A.Br.003/004 (Fig 5), corresponding to the Trans-Eurasian (TEA) lineage, which has a global distribution, including South America [62,70]. This lineage has been previously documented in neighbouring countries such as Chile, Argentina, and Bolivia [63], suggesting widespread regional dissemination. Strains isolated in Lima in 2005 formed a distinct cluster, indicating a possible localized outbreak, whereas isolates from Piura and Lambayeque displayed greater genetic diversity (Table 1), consistent with sporadic or localized transmission events. Altogether, these findings highlight the genetic heterogeneity of *B. anthracis* in Peru and emphasize its relevance to epidemiological surveillance and anthrax control efforts.

## Limitations

Our work has some limitations that should be acknowledged. Although whole-genome sequencing allowed us to identify key antimicrobial resistance and virulence genes, we did not perform phenotypic antimicrobial susceptibility testing to experimentally validate these predictions. As a result, our conclusions are based on genomic analyses, and future studies integrating genomic data with phenotypic assays will be important to confirm the functional expression of these traits. In addition, the limited availability of publicly accessible *Bacillus anthracis* genomes from South America restricted broader regional phylogenetic comparisons. Expanding genomic surveillance efforts in the region will be essential to place Peruvian isolates into a more comprehensive epidemiological context.

## Conclusions

This study represents the first genomic characterization of *B. anthracis* strains isolated in Peru between 2005 and 2017. Phylogenetic analysis showed that all isolates belonged to lineage A, subgroup A.Br.003/004, with limited overall genetic diversity. A total of 26 virulence genes with high identity (>94%) were identified, including *inhA1* and *inhA2* in all strains, along with a high prevalence of *alo*, *hlyII*, and *nhe*, underscoring the broad spectrum of factors involved in pathogenesis. The presence of pXO1 and pXO2 plasmids—critical components of the bacterium's virulence potential—was also confirmed.

Regarding antimicrobial resistance, eight genes were identified with identity and coverage above 82%, notably *bla* and *bla2*, present in all strains and associated with resistance to β-lactams and carbapenems. These findings, generated through whole genome sequencing, reinforce the importance of implementing genomic surveillance of the pathogen in endemic settings like Peru, in order to improve strategies for controlling, preventing, and responding to anthrax outbreaks.

## Supporting information

**S1 Table. Summary of core genome SNPs counts and variants for *Bacillus anthracis* isolates.** This table details the SNP counts and variant statistics for all isolates included in the study.
(XLSX)

## Acknowledgments

This research is partly derived from the master's thesis of Eissen E. Guerrero-Seminario, conducted at the Postgraduate Unit of the Faculty of Biological Sciences, Universidad Nacional Mayor de San Marcos, Lima, Peru. Ruth García-de-la-Guarda and Ever F. Córdova-Diaz served as thesis advisors.

## Author contributions

**Conceptualization:** Eissen E. Guerrero-Seminario, Abraham Espinoza-Culupú, Ruth García-de-la-Guarda.

**Formal analysis:** Eissen E. Guerrero-Seminario, Abraham Espinoza-Culupú, Ruth García-de-la-Guarda.

**Investigation:** Eissen E. Guerrero-Seminario.

**Methodology:** Eissen E. Guerrero-Seminario, Abraham Espinoza-Culupú, Patricia García-Vara, John Calderón-Escalante, Dana González-Quispe, Beitzy Cubas-Yalle, Lourdes Balda Juárez.

**Resources:** John Calderón-Escalante.

**Supervision:** Ever F. Córdova-Diaz, Ruth García-de-la-Guarda.

**Validation:** Eissen E. Guerrero-Seminario, M. Angelica Delgado-Baldeon.

**Writing – original draft:** Eissen E. Guerrero-Seminario, Abraham Espinoza-Culupú.

**Writing – review & editing:** Eissen E. Guerrero-Seminario, Abraham Espinoza-Culupú, Ruth García-de-la-Guarda.

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
