## [Decision Letter · Decision Letter 0]

10 Dec 2025

Response to Reviewers
Revised Manuscript with Track Changes
Manuscript

Shaden Kamhawi

co-Editor-in-Chief

Paul Brindley

co-Editor-in-Chief

**Additional Editor Comments:**
**Journal Requirements:**

- ® on page: 3 and 4.

4) Tables should not be uploaded as individual files. Please remove these files and include the Tables in your manuscript file as editable, cell-based objects. For more information about how to format tables, see our guidelines:

https://journals.plos.org/plosntds/s/tables

Potential Copyright Issues:

- Figure 1. Please (a) provide a direct link to the base layer of the map (i.e., the country or region border shape) and ensure this is also included in the figure legend; and (b) provide a link to the terms of use / license information for the base layer image or shapefile. We cannot publish proprietary or copyrighted maps (e.g. Google Maps, Mapquest) and the terms of use for your map base layer must be compatible with our CC BY 4.0 license.

**Reviewers' comments:**

**Key Review Criteria Required for Acceptance?**

**Methods:**

-Are the objectives of the study clearly articulated with a clear testable hypothesis stated?

-Is the study design appropriate to address the stated objectives?

-Is the population clearly described and appropriate for the hypothesis being tested?

-Is the sample size sufficient to ensure adequate power to address the hypothesis being tested?

-Were correct statistical analysis used to support conclusions?

-Are there concerns about ethical or regulatory requirements being met?

Reviewer #1: The work “Genomic Characterization of Clinical and Environmental Isolates of Bacillus anthracis in Peru” is important for understanding the epidemiology of the strains circulating in the country, also serving as a geographic database, as there are few Bacillus anthracis genomes sequenced in South America.

For publication, some aspects need to be improved and clarified:

1- Introduction: The authors presented data from publications on the number of cases and outbreaks. This information could be improved with data from official government agencies with the number of current cases in humans and animals. What is the current number of anthrax cases in Peru?

2. Methods: For the phylogenetic analysis, what was the criterion for choosing the 35 strains for genome comparison? Why were no strains from South America included? It would be interesting to compare them with the genomes of isolates from South America, in order to improve the phylogenetic analysis.

Reviewer #2: The objectives are clearly articulated and the study design is appropriate. The B.anthracis outbreak samples were clear describe from their origin and the year of isolation. Being a whole genome sequencing study, there was no requirement to calculate a sample size and the relevant analysis was DNA extraction and amplification, Whole genome sequencing and Bioinformatics analysis. This was described well in the methods section.

Reviewer #3: 1) Yes, the objectives are clearly stated.

2) Yes, the study design is sufficiently appropriate.

3) Yes, the population is clearly described; it comprises a set of B. anthracis isolated from two major parts of Peru.

4) The authors have stated no specific hypothesis, and in my opinion none is needed for this study. The study is a survey of B. anthracis collected in Peru using the whole genome sequencing approach, and as such, I think it can stand on its own without a hypothesis.

5) Yes, the extent of statistical analysis needed is appropriate.

6) There seem to be no ethical or regulatory concerns; in any case, if there are any Peruvian, WHO, or US regulations applicable to this study, then I am not aware. In my opinion there are none, as long as the bacterial strains are properly safeguarded.

**Results:**

-Does the analysis presented match the analysis plan?

-Are the results clearly and completely presented?

-Are the figures (Tables, Images) of sufficient quality for clarity?

Reviewer #1: Results: Although described in the methodology, the SNP results were not demonstrated. These results could help in a more specific phylogenetic analysis of the clade.

Reviewer #2: The results are clear and illustrative and are consistent with the described methods. The whole genome sequencing, phylogenic analysis and the analysis of the plasmids was well described and illustrated. There is also clarity on the antimicrobial resistant genes identified. The tables, images and figures are clear and exhaustive.

Reviewer #3: Yes, to all three questions. However, citing of Table 1 and the four figures within the narrative texts of Results and Discussion are scant, as they appear only once each. Figure 4 is not cited anywhere in the textual narrative.

I suggest the authors cite the table and figures more frequently; once is not enough.

**Conclusions:**

-Are the conclusions supported by the data presented?

-Are the limitations of analysis clearly described?

-Do the authors discuss how these data can be helpful to advance our understanding of the topic under study?

-Is public health relevance addressed?

Reviewer #1: The delimitations of the analyses were not clearly described.

Reviewer #2: The discussion and conclusion is consistent with the finding on the genomic sequencing and this study adds to the body of knowledge and will assist in genomic surveillance during outbreaks and in general practice. The author could have been more overt on any limitations that may arise from the study. This study will guide in the prevention and management of Anthrax in Peru and contributes to the understanding of genotypic expression of B.anthracis in the South American region and globally

Reviewer #3: Yes, to all four questions.

**Editorial and Data Presentation Modifications?**

Reviewer #1: (No Response)

Reviewer #2: Accept

Reviewer #3: 1) Italicize bacterial names wherever they occur.

2) Cite the table and four figures more often; it would help the readers understand and pin the textual narratives with the data presented in the form of figures.

3) Because the authors have the complete genomic sequences of all B. anthracis samples, they should mention likely functionality of the ABR and virulence genes they mention. I think it should be feasible to determine from the gene sequences whether they are functional. If the sequences match the known wild type sequences, including those of the noncoding regions, then I think the authors can reasonable say that the genes and their products are likely functional. They may even add a table that lists names of the key genes in one column and any sequence deviations they found in the next column. They could add some relevant comments as footnotes regarding each sequence deviation and its functional implications. Doing so in my opinion would make their manuscript stronger.

**Summary and General Comments**

Reviewer #1: Suggestion for improving the complete characterization of the lineages that could assist epidemiological surveillance, since there is a different cluster and the detection of several resistance genes:

Inclusion of Antimicrobial susceptibility testing for comparison with the results found in the analyses for AMR genes to verify their functionality.

Expand the phylogeny analyses by comparing with genomes from South America: Argentina, Chile, Bolivia and Brazil (where a subclade was described)

Include SNP analyses.

With the inclusion of these analyses, the discussion and conclusion can be improved and more data correlated with phylogeny and resistance.

Reviewer #2: The overall study is of sound quality as it is the first study to characterize the genome of B.anthracis in Peru from 2005 – 2017 from outbreak samples. The genome studies have shown high homogeneity of the samples, presence of antimicrobial resistance genes and also analysed plasmids PX01 and PX02 showing virulence potential of the bacterial. It is a very important study as it sheds light on the source and potential distribution of this bacterial that has been used for bio terrorism and biological weaponry.

Reviewer #3: The authors report a whole genome survey of B. anthracis samples collected in two regions of Peru. Their method was whole genome sequencing. The study is important, and it adds to the vast repertoire of B. anthracis sequencing data. The manuscript is nicely written, and it neither exaggerates, nor diminishes the significance of the findings.

My major suggestion, which I mentioned above is that they should address whether, based on the sequences, any or all of these genes have the sequences that result in their products. If they found sequence deviations, then they should appropriately comment on those.

PLOS authors have the option to publish the peer review history of their article (what does this mean? ). If published, this will include your full peer review and any attached files.

**Do you want your identity to be public for this peer review?** For information about this choice, including consent withdrawal, please see our Privacy Policy .

Reviewer #1: No

Reviewer #2: No

Reviewer #3: No

**Figure resubmission:**

**Reproducibility:** To enhance the reproducibility of your results, we recommend that authors of applicable studies deposit laboratory protocols in protocols.io, where a protocol can be assigned its own identifier (DOI) such that it can be cited independently in the future. Additionally, PLOS ONE offers an option to publish peer-reviewed clinical study protocols. Read more information on sharing protocols at https://plos.org/protocols?utm_medium=editorial-email&utm_source=authorletters&utm_campaign=protocols

---

## [Editor Report · Decision Letter 1]

15 Jan 2026

Dear Dr. Garcia-de-la-Guarda,

We are pleased to inform you that your manuscript 'Genomic Characterization of Clinical and Environmental Isolates of Bacillus anthracis in Peru' has been provisionally accepted for publication in PLOS Neglected Tropical Diseases.

Best regards,

Jeffrey H. Withey, Ph.D.

Academic Editor

Ana LTO Nascimento

Section Editor

Shaden Kamhawi

co-Editor-in-Chief

Paul Brindley

co-Editor-in-Chief

---

## [Editor Report · Acceptance letter]

Dear Professor García-de-la-Guarda,

We are delighted to inform you that your manuscript, "Genomic Characterization of Clinical and Environmental Isolates of Bacillus anthracis in Peru," has been formally accepted for publication in PLOS Neglected Tropical Diseases.

Best regards,

Shaden Kamhawi

co-Editor-in-Chief

Paul Brindley

co-Editor-in-Chief
